# Influence of Native *S. cerevisiae* Strains on the Final Characteristics of "Pago" Garnacha Wines from East Spain

Carmen Berbegal [1,2], Lucía Polo [1], Victoria Lizama [3,4], Inmaculada Álvarez [3,4], Sergi Ferrer [1,2], Isabel Pardo [1,2] and Mª José García-Esparza [3,4,*]

[1] ENOLAB, Institut Universitari de Biotecnología i Biomedicina (BIOTECMED). c/Dr. Moliner 50, 46100 Burjassot, Spain
[2] Departament de Microbiologia i Ecología, Universitat de València. c/Dr. Moliner 50, 46100 Burjassot, Spain
[3] Instituto de Ingeniería de Alimentos para el Desarrollo, Universitat Politècnica de València, Camino de Vera s.n., 46022 Valencia, Spain
[4] Departamento de Tecnología de los Alimentos, Universitat Politècnica de València, Camino de Vera s.n., 46022 Valencia, Spain
* Correspondence: mesparza@tal.upv.es

**Abstract:** This work studies the variability of the *Saccharomyces cerevisiae* present during the spontaneous fermentation of Garnacha grapes' musts from a "Pago" winery from the east of Spain. The parameters used to select yeast are those related to growth, fermentative behaviour, and the influence on the wine's aroma and polyphenolic composition. Yeast identification was performed by ITS analysis and typed by Hinfl mDNA restriction profile analysis. Growth and metabolic characteristics of the isolates were determined by laboratory-scale fermentations of sterile Garnacha must, and the composition of the polyphenolic and the volatile compounds, and the sensory attributes of the small-scale produced red wines were determined. Ten *S. cerevisiae* strains were isolated and characterized. Overall, strain 22H quickly grew, produced wines with moderate ethanol concentrations and low volatile acidity, and obtained the highest colour and aroma scores, plus a high score for sensory attributes.

**Keywords:** *Saccharomyces cerevisiae*; volatile compounds; colour parameters; yeast characterisation

## 1. Introduction

Vineyard soil, climate, grape variety, viticultural and oenological practices determine the microbial composition that will participate in alcoholic fermentation (AF). The "Terroir" term refers to the diversity of the strains and interactions associated with regional variations, conferring each wine's distinction [1].

Bokulich, Collins, Masarweh, Allen, Heymann, Ebeler, and Mills [1] have shown that grape microbiota and wine metabolic profiles distinguish wine regions, although the degree of the relationship with wine chemical composition is still not sufficiently clear. "Pago" is a well-defined geographical area with very specific climatic and soil composition characteristics that differentiate it from its surroundings, where wines with unique characteristics and qualities are obtained [2].

From the microbiological point of view, winemaking is a complex process in which a plethora of microorganisms play roles of diverse importance. Thus, we can find many species of filamentous fungi, yeasts, and lactic acid and acetic acid bacteria in freshly crushed grapes. The addition of $SO_2$ as an antioxidant agent, the evolution of physicochemical conditions, and the interaction among microorganisms during AF result in the initial microbial diversity drastically decreasing [3–5]. During the first hours of spontaneous AF, the concentrations of non-*Saccharomyces* species are usually higher than *Saccharomyces* species [6], but *Saccharomyces cerevisiae* is the yeast that prevails later and is responsible for the transformation of sugar into ethanol [7]. Several years ago, non-*Saccharomyces*

species development was considered negative, but today the negative consideration of these species is now positive because non-*Saccharomyces* populations can improve wines' typicity [4].

However, *S. cerevisiae* dominates AF [8]. The importance of the role of *S. cerevisiae* and its influence on the chemical composition of wines is proven by the study of fermentations carried out by different strains of this species and by using the same must variety, which show marked variability in the generated compounds. This fact indicates that one of the factors influencing the final wine quality is the interaction between different *S. cerevisiae* strains and grape must composition [9–12]. Thus, wines of different compositions (variations in glycerol, ethanol, or acetic acid production) [13], and with differing sensory and aromatic profiles [9] can be obtained using distinct *S. cerevisiae* strains.

Musts can be left to either ferment with the natural yeasts present in grapes or can be inoculated with selected commercial yeast strains. Both strategies have positive and negative connotations. Spontaneous fermentations with native yeasts are becoming increasingly less frequent in winemaking because there is no guarantee that this will take place properly; however, this strategy contributes to wines' typicity and originality. Commercial yeasts ensure good fermentation kinetics and the production of beneficial compounds for wine quality, but produce wines with more similar characteristics [9]. Presently, there is a tendency to make wines with selected indigenous yeasts of each wine-growing area or grape variety to combine reliability and wine differentiation. The selected native yeasts, in addition to adequate fermentative characteristics (high fermentative power, good ability to exhaust sugars, high growth rate, ability to produce compounds to improve sensory quality, etc.), are better adapted to grow in the must from which they have been isolated than in commercial yeasts [4].

A yeast selection programme implies the isolation of many yeasts directly from the grapes from a given vineyard or wine-growing area or from freshly crushed grape musts [3], as well as their oenological characterisation according to criteria which ensure good technological and oenological performance of the selected yeasts [14]. Several authors have shown that fermentation with different *S. cerevisiae* strains results in wines with different chemical compositions and distinct polyphenolic and aromatic profiles [15–18].

Many studies have reported the influence that different yeast strains have on the polyphenolic composition (total concentration, degree of polymerisation, type of polyphenol, colour stability, etc.), and on the aromatic compounds and overall wine complexity [19–24].

This work studies the variability of the *S. cerevisiae* present in the spontaneous fermentation of Garnacha grapes musts of a "Pago" wine category from the Utiel-Requena region of eastern Spain. The main objective is to select the most suitable strains to obtain quality wines. The parameters used to select yeast are those related to growth, fermentative behaviour, and the influence on wines' aroma and polyphenolic composition.

## 2. Material and Methods

### 2.1. Winery Characteristics

The winery from which the samples were obtained is located in the Utiel-Requena region, 70 km from Valencia (eastern Spain). The winery has an 80-hectare vineyard, of which 7 hectares are used to grow the Garnacha variety. This winery produces 500,000 kg of grapes a year, of which 50,000 kg correspond to the Garnacha variety. The wines produced with these grapes fall into the "Pago" wines category, and the AF of these wines is exclusively performed by indigenous yeasts.

### 2.2. Physico-Chemical Characteristics of Garnacha Grape Musts

Grape must had a density of 20.8° Brix, a probable alcoholic degree of 11.90% ($v/v$), a titratable acidity of 5.22 g/L (expressed as tartaric acid), and a pH of 3.59.

### 2.3. Counts and Isolation of Native Yeasts from Industrial Fermentations

Samples were obtained from the spontaneous fermentation of a 10,000 L tank of Garnacha grape must. Yeasts were isolated at three times: in must before AF, halfway through AF (HAF), and at the end of AF (FAF). The appropriately diluted samples (in triplicate) were spread on yeast extract, peptone, and dextrose (YPD) plates, and incubated at 28 °C for 48–72 h. The concentration of culturable cells was estimated by counting the colonies that appeared on the YPD plates, expressed as colony-forming units per millilitre (CFU/mL). The colonies with different morphologies corresponding to the grape must samples were isolated by streaking on YPD plates. Ten colonies (chosen randomly) from the YPD plates corresponding to the HAF and FAF samples were also individually streaked on YPD plates. Isolates were preserved frozen at −20 °C in a 50% mixture of YPD-grown cells and 30% glycerol solution until use.

### 2.4. Identification and Typing of Yeasts

The identification of yeast isolates was performed by an Internal Transcribed Sequences analysis (considering both length and sequence). The ITS1 and ITS4 primers described by Esteve-Zarzoso et al. [25] were used to amplify a region of the rRNA gene repeat unit, which includes the internal transcribed spacers (ITS1 and ITS2) and the 5.8S rRNA gene. The conditions applied to amplify the ribosomal region were similar to those described by Esteve-Zarzoso, Belloch, Uruburu, and Querol [25], with some modifications: 50 µL reaction volume containing 5 µL Taq buffer, 1 µM each primer, 0.1 µMgCl$_2$, 0.01 U EuroTaq Taq Polymerase EuroClone (Milan), 0.8 µM dNTP from Roche, 25 µL of cell suspension (1 colony in 25 µL of sterile Milli-Q water), and Milli-Q water up to 50 µL. Preliminary yeast classification was based on ITS fragment lengths was performed by employing the public database https://www.yeast-id.org/ (accessed on 25 October 2021) [25]. When the length analysis did not discriminate between species, ITS sequencing was performed at the Servei Central de Suport a la Investigació Experimental (SCSIE) of the Universitat de València, and identity was determined by BLAST.

Yeast isolates were typed by a mitochondrial DNA (mDNA) restriction pattern analysis with HinfI from Roche (Barcelona, Spain). The procedure was as described by Querol et al. [26] with slight modifications. Modifications were as follows: lowering the concentrations of sorbitol and sodium dodecyl sulphate to 0.9 mol/L and 0.26%, respectively, instead of 1 mol/L and 1%; Zymolyase 20 T solution (United States Biological, Salem, MA, USA) was used at a final concentration of 0.07 mg/mL; the incubations at 65 °C and on ice were performed for 30 min and 5 min, respectively; cell debris centrifugation was increased from 5 to 10 min; finally, purified DNA was dissolved in 50 mL Tris-EDTA (pH 8). HinfI restriction digestion was carried out using 10 µL of extracted DNA, 2 µL of reaction buffer R, 1 µL of HinfI (10 U/µL; Sigma-Aldrich, St. Louis, MO, USA), 1 µL RNAase (4 mg/mL; Roche, Barcelona, Spain), and 6 µL Milli-Q water. The reaction mixture was incubated overnight at 37 °C.

Restriction fragments were resolved by gel electrophoresis on 0.8% agarose gel in 0.5× Tris borate EDTA buffer at 20 V for 16 h before being stained with ethidium bromide. Gels were digitalised and HinfI restriction profiles were compared to one another with the BioNumerics 5 software (Applied Maths, Kortrijk, Belgium) by following the grouping method "The Unweighted Pair Group Method with Arithmetic Mean" (UPGMA) and "Pearson's Product-Moment Coefficient". The isolates with the same mDNA HinfI profile were considered to be the same strain. One isolate of each strain was chosen as being representative of the group and characterised as described below.

### 2.5. S. cerevisiae Strain Characterisation

Yeast characteristics were determined in the same Garnacha must from which yeasts were isolated. The parameters used to evaluate strains were growth-related [growth kinetics, growth rate, 7-day cell concentration (7dCC), and Area Under the Curve (AUC)],

and metabolism-related (glucose and fructose consumption kinetics, ethanol, glycerol and acetic acid production kinetics, ethanol yields) parameters.

Garnacha must was centrifuged in a Beckman coulter Avanti J-E centrifuge (Brea, CA, USA) at 17,696 g and 4 °C for 40 min to settle solids and native microorganisms. After recovering the supernatant, it was sterilised by adding 0.25 g/L of Velcorin® (Lanxess, Cologne, Germany), which was left to act at room temperature for 5–6 h before yeast inoculation. Velcorin® (dimethyl dicarbonate or DMDC) penetrates the cell and deactivates enzymes, leading to the destruction of the microorganisms. Yeast isolates were grown in YPD broth at 28 °C for 48 h. Yeast concentrations were determined by microscopic counting in a Thoma chamber (Llinars del Vallès, Barcelona, Spain) and by inoculating YPD plates. The YPD-precultured yeasts were inoculated in Garnacha must at a final concentration of $2 \times 10^5$ cells/mL and incubated at 28 °C for 14 days. Samples were taken on days 0, 1, 3, 7, and 14. Fermentations were performed in triplicate. Yeast growth kinetics were determined by culturable cell (CFU) counts per mL on YDP plates (UFC/mL). The $\mu_{max}$ was determined as the increase in UFC/mL during the first 24 h of fermentation; the 7dCC was the CFU/mL value found at this fermentation time; and the AUC was the measure corresponding to the whole two-dimensional area underneath the entire growth curve [27], when considering the entire kinetics curve.

The kinetics of glucose, fructose consumption and ethanol, glycerol, and acetic acid production were determined by measuring their concentrations during fermentation (at 1, 3, 7, and 14 days) by high-performance liquid chromatography (HPLC) following the procedure described by Frayne [28]. Both sugars' consumption and ethanol concentrations on day 14 were used to compare the strains' ethanol yield. One must sample (in triplicate) before yeast inoculation (time 0) was also analysed.

### 2.6. Small-Scale Red Wine Fermentation

The influence of yeast strains on the polyphenol composition, aroma characteristics, and sensorial attributes of Garnacha wines was determined by small-scale fermentations in Garnacha grape must with SO₂ g/L added, as described below.

Grapes were harvested in 10-kg boxes, and were manually destemmed and frozen at the *Instituto de Ingeniería de Alimentos para el Desarrollo* of the *Universitat Politècnica de València*. Grapes were thawed on the day before the winemaking process began. Then, 1.6 kg of crushed grapes were placed in 2-kg jars, and 200 mg/kg Velcorin® were added to eliminate the indigenous flora of grapes before adding 50 mg/kg potassium bisulphite. After 24 h, the different *S. cerevisiae* yeast strains were inoculated at a rate of $2 \times 10^5$ cells/mL. Small-scale vinification was carried out in triplicate. AF was conducted at 25–26 °C, and lasted approximately 10 days. During fermentation, punching down was performed to favour the extraction of polyphenolic compounds. Fermentation was monitored by determining temperature and density to check the fermentation kinetics and to verify the absence of fermentation arrests. Fermentation was considered complete at a density of 992–993 g/cm³ and when the reducing sugars of wines were between 1 and 2 g/L. Having completed AF, the commercial liquid starter *Oenococcus oeni* OE104 (Agrovin, Alcazar de San Juan, Spain) was added at a dose of 0.13 mL/L of wine. Wines completed malolactic fermentation (MLF) between 15 and 20 days after inoculation. MLF development was monitored by paper chromatography and once completed, wines were racked and sulphited with potassium bisulphite to obtain a concentration of 30 mg/L of free sulphur dioxide. Wines were bottled in 500 mL bottles and left to settle for 1–2 months at a temperature between 16 °C and 18 °C before analysing their chemical composition and sensorial characteristics.

### 2.7. Analytical Methods

The common parameters (density, ethanol, pH, titratable acidity (TA), volatile acidity) in musts and wines were determined according to the Official Methods of the European Commission [29]. Total soluble solids (TSS) (°Brix) were determined by refractometry and reducing sugars by the Fehling method [30]. The rate at which sugars were converted into

ethanol was calculated as the sugars consumed (g/L) divided by ethanol produced (%, *v/v*).

Phenolic wine composition was determined in a JASCO V-530 UV-Visible spectrophotometer and a JASCO MD2010 Plus HPLC coupled with a diode array detector (DAD) (JASCO LC-Net II/ADC, Tokyo, Japan). All the measurements were taken in triplicate. Colour intensity, hue, TPI (Total Polyphenols Index), and the Gelatin Index (astringency) were estimated by the methods described by Glories [31]. Condensed tannins were determined by the method by Peynaud et al. [32]. The Folin–Ciocalteu assay was run according to Singleton and Rossi [33]. The method reported by Boulton [34] was followed to analyse the contribution of the co-pigmented, free, and polymeric anthocyanins to total wine colour. The Ribéreau-Gayon and Stonestreet [35] method was followed to determine the bisulphite non-bleached anthocyanins (coloured anthocyanins). Catechins were quantified by the method of Sun et al. [36]. Total condensed tannins were assessed after heat transformation into anthocyanidins in an acidic medium [32]. The polyvinylpolypyrrolidone (anthocyanin–tannin complexes) and dimethylamino cinnamaldehyde (DMACH; degree of tannin polymerisation) indices were calculated according to Vivas et al. [37].

The individual anthocyanins compounds were quantified by HPLC via the method of Boido et al. [38]. Total anthocyanins were calculated as the sum of glucoside anthocyanins and acylated anthocyanins. After centrifugation and filtration, wine samples were injected directly into the HPLC (20 µL). Separation was carried out in a Gemini NX (Phenomenex, Torrance, CA, USA) 5 µm, 250 mm × 4.6 mm i.d. column at 40 °C. Solvents were 0.1% trifluoroacetic acid (A) and acetonitrile (B). The elution gradient was as follows: 100% A (min 0); 90% A + 10% B (min 5); 85% A + 15% B (min 20); 82% A + 18% B (min 25); 65% A + 35% B (min 30). Individual chromatograms were extracted at 520 nm. For quantification, calibration curves were obtained with a commercially available standard: malvidin-3-glucoside (Sigma-Aldrich, St Louis, MO, USA). Anthocyanins content was calculated on the basis of the calibration curves of authentic malvidin-3-glucoside ($y = 236{,}316x - 166{,}569$, $R^2 = 0.9994$).

Volatile compounds were analysed following the procedure proposed by Ortega et al. [39] with slight modifications. A volume of 2.7 mL of samples was transferred to a 10 mL screw-capped centrifuge tube containing 4.05 g ammonium sulphate (Panreac, Barcelona) to which the following compounds were added: 6.3 mL miliQ water (Panreac), 20 µL standard internal solution (2-butanol, 4-methyl-2-pentanol, and 2-octanol from Aldrich, at 140 µg/mL each, in absolute ethanol from LiChrosolv-Merck (Darmstadt, Germany), and 0.25 mL dichloromethane (LiChrosolv-Merck). The tube was shaken mechanically for 120 min and later centrifuged at 4000 rpm for 15 min. The dichloromethane phase was recovered with a 0.5 mL syringe, transferred to the autosampler vial, and analysed. The chromatographic analysis was carried out in a HP-6890 equipped with a ZB-Wax plus column (60 m × 0.25 mm × 0.25 µm) from Phenomenex. The column temperature was initially set at 40 °C and was left at this temperature for 5 min, raised to 102 °C at a rate of 4 °C/min; to 112 °C at a rate of 2 °C/min; to 125 °C at a rate of 3 °C/min, and maintained at this temperature for 5 min before being raised to 160 °C at a rate of 3 °C/min; to 200 °C at a rate of 6 °C/min, and then left at this temperature for 30 min. The carrier gas was helium, which was fluxed at a rate of 3 mL/min. The injection was carried out in the split mode 1:20 (injection volume 2 µL) with a flame ionisation detector (FID).

### 2.8. Statistical Analysis

Data were analysed for statistical significance by a one-way analysis of variance (ANOVA, $p \leq 0.05$), and the bivariate correlations between the analysed variables were determined by Pearson's correlation coefficient.

All the analyses were submitted in triplicate for each fermentation replicate. The results are expressed as mean values ± SD. To determine if yeast significantly affected the physico-chemical, phenolic compounds and volatile aromatic composition of wines, a simple ANOVA analysis was run by taking a 95% confidence level. The existence of a

significant difference between yeasts was studied for each parameter. The Statgraphics Centurion XVI software (Statgraphics Technologies, The Plains, VA, USA) was used for this analysis.

To simplify the results, a principal component analysis (PCA) and orthogonal projections to the latent structures discriminant analysis were performed with SIMCA, version 10 (MKS Umetrics, Malmo, Sweden). The PCA was used to identify the main factors that explained most of the variance observed from a much larger number of manifest variables.

*2.9. Sensory Analysis*

The sensory analysis of the fermented wines with the different *S. cerevisiae* strains was performed by a panel of 10 expert tasters, who had previously been submitted to selection and training [40]. Tasting was carried out under standardised conditions in a tasting room with standardised cabins [41]. Initially, triangular tests were undertaken according to Standard ISO 4120 [42] for the three wine repetitions to determine whether there were sensorial differences between them, and to then average the values obtained with the sensory analysis. The descriptive and quantitative sensory analysis [43] was performed during a single session to avoid the influence of the different physical conditions while the evaluators tasted wines.

## 3. Results and Discussion

*3.1. Counts, Isolation, Identification, and Typing of Yeast Counts from Industrial Garnacha Fermentations*

The yeast concentrations in Garnacha were $1.8 \times 10^4 \pm 1.2 \times 10^3$ CFU/mL in grape must before AF, and $2.6 \times 10^7 \pm 1.8 \times 10^6$ and $8.4 \times 10^6 \pm 8 \times 10^5$ CFU/mL at HAF and at FAF, respectively. Thirty-two yeast isolates were recovered at these three fermentation times. ITS fragment lengths of 390, 450, 620, and 750 bp were found but, as various yeast species share the same ITS lengths, it was necessary to sequence the fragment to identify our isolates. The isolates exhibiting a fragment of 390 bp were identified as *Metschnikowia pulcherrima* (26% relative frequency); isolates with a 450 bp ITS resulted to be *Issatchenkia terricola* (1.7% relative frequency); the isolate showing a 620 bp ITS fragment was identified as *Wickerhamomyces anomalus* (0.6% relative frequency); some isolates exhibiting a 750 bp ITS were identified as *Hanseniaspora guilliermondii* (55% relative frequency), and *Hanseniaspora opuntiae* (16.6% relative frequency); and isolates with a 850 bp ITS resulted to be *S. cerevisiae* (10%). At HAF, the most abundant species was *S. cerevisiae* (86.3%), but small *H. guilliermondii* populations (13.7%) remained. Only *S. cerevisiae* (100%) was found at the end of AF (FAF) (Table 1).

**Table 1.** *S. cerevisiae* yeasts isolated from industrial Garnacha Pago wine: isolate name, origin, mDNA HinfI profile, and representative profile isolate. The right column describes the isolate that represents each mDNA pattern. GM: Grape must; HAF: Halfway (middle) through alcoholic fermentation; FAF: End of alcoholic fermentation.

| Isolates | Isolated From | Profile Number | Representative Profile Isolate |
|---|---|---|---|
| 17A | GM | | |
| 22I | HAF | I | 17A |
| 17B, 22A, 22B, 22C, 22D, 22E, 22H, 22J, 23A, 23B, 23C, 23E | HAF | | |
| 38D, 38F, 38H, 39A, 39D | FAF | II | 22H |
| 22F, 22G | HAF | III | 22F |
| 24B | HAF | | |
| 39C | FAF | IV | 39C |
| 38A, 38G, 38J, 38I | FAF | V | 38A |
| 38B | FAF | VI | 38B |
| 38C | FAF | VII | 38C |
| 38E | FAF | VIII | 38E |
| 39B | FAF | IX | 39B |
| 39F | FAF | X | 39F |

Spanish spontaneous Garnacha fermentations were previously analysed by Portillo and Mas [44], and Padilla et al. [45] by high throughput sequencing (HTS) and plating isolation, respectively. By HTS, Portillo and Mas [44] found yeast belonging to the genera *Hanseniaspora* and *Issatchenkia* in grape must, whereas Padilla, García-Fernández, González, Izidoro, Esteve-Zarzoso, Beltran, and Mas [45] observed *H. uvarum* and *Issatchenkia terricola* in some of the four analysed Grenache (Garnacha) grape musts. The presence of the *Hanseniaspora* species as *H. uvarum* or *H. guilliermondii* (or their teleomorphs *Kloeckera apiculata* and *Kloeckera apis),* as well as *M. pulcherrima,* is quite common in the grape must of many varieties [46–49]. The isolation of *H. opuntiae*, *I. terricola*, and *W. anomalus* (formerly *Pichia anomala*, *Hansenula anomala*, *Candida pelliculosa*) is less frequent. *I. terricola* has been found in the grape must or at early fermentative times of Macabeo, Vermentino, and Viognier white wine fermentations [47].

After applying the UPGMA grouping method and Pearson´s product-moment coefficient to the mDNA HinfI profiles of the 32 *S. cerevisiae* isolates, 10 groups of profiles were detected at the 86% cut-off level. They were labelled as roman numerals I to X (Figure 1). It was assumed that each different profile was a distinct strain. In GM, only one isolate was found (17A) to exhibit the mDNA profile I (Table 1). Sixteen isolates were recovered at HAF, which were grouped as four different profiles: I, II, III, and IV (Table 1). Fifteen isolates were obtained at FAF and were grouped as eight profiles (strains) II, IV, V, VI, VII VIII, IX, and X (Table 1). The strain with profile I remained until HAF, but was a minority and then disappeared. The isolates with profiles II and IV (strains) appeared at HAF and remained until FAF, whereas six new different profiles (strains) (profiles V, VI, VII VIII, IX, and X) were exclusively noted at FAF. Of the 16 isolates recovered at HAF, 12 belonged to profile II, which showed the highest frequency at that time (75%), whereas the frequency of profiles I, II, and IV was low (6.3%, 12.5%, and 6.3%, respectively). At FAF, profiles (strains) II and V were the most abundant with 33.3% and 26.7%, respectively, whereas six other profiles were found at low frequency (6.7% each). The low *S. cerevisiae* diversity observed in GM was possibly the consequence of its low concentration in the freshly obtained must, and was above the detection threshold of our assay; the low frequency of *S. cerevisiae* in both grapes and grape juice has been reported by several authors [45,50,51]. It would seem that *S. cerevisiae* diversity increased as AF progressed. The same trend in diversity evolution, strain succession, and the dominance of one strain or two during fermentation have been reported by Le Jeune et al. [52] and Ribéreau-Gayon, Dubourdieu, Donèche, and Lonvaud [6] in white Alsace and red Bordeaux fermentations, respectively. We observed an increase in the number of *S. cerevisiae* strains as AF progressed, unlike that reported by other authors. This fact is possibly due to different yeast interactions, grape must characteristics, and yeast metabolic abilities. Thus, differences in fermentative power, ethanol sensitivity, $SO_2$ production and resistance to it, the secretion of killer-like compounds, and the nutrient requirements of *S. cerevisiae* strains drive the dynamics of the *S. cerevisiae* populations [53,54].

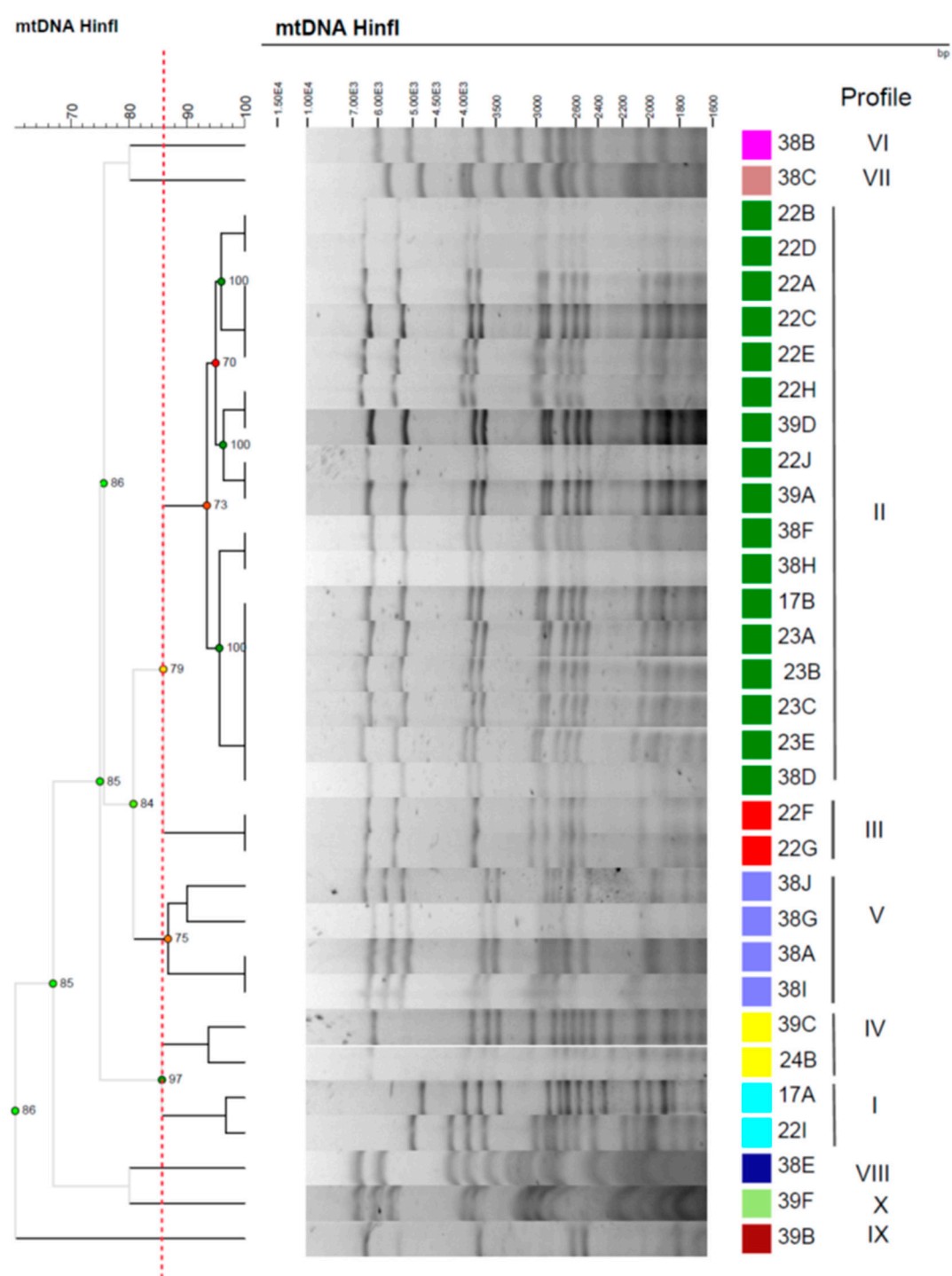

**Figure 1.** Dendrogram based on the similarities of the mDNA HinfI restriction profiles built using Pearson's product-moment correlation coefficient and the unweighted pair group method with arithmetic mean (UPGMA). Cut-off level set at 86% similarity.

*3.2. S. cerevisiae Yeast Characterisation*

The characteristics of the growth-related and metabolism-related parameters of the 10 *S. cerevisiae* strains were determined in the Garnacha grape must from which they were isolated.

The growth kinetics of the 10 strains are described in Figure S1. As the comparison between the growth kinetics of the different yeasts was not easy, the growth-related param-

eters, such as AUC, $\mu_{max}$, and 7dCC are also reported in Figure 2A–C, respectively. AUC translates growth kinetics into a numerical value. Figure 2A shows important differences in the AUC values among the yeast strains. Hence, the strains with higher values were 22F, 38C, 39B, 39C, and 39F, whereas those with lower values were 17A, 22H, 38A, and 38E; these last strains only showed significant differences with strain 22F. Different AUC values reflect differences in lag, exponential, stationary, and death phase duration [27]. In relation to $\mu_{max}$, the faster growing strains were 38A, 38B, 22H, 39B, and 39C, whereas strains 22F, 39B, and 39F grew more slowly (Figure 2B); only strains 22F and both 38A and 38B showed significant differences in this parameter. Strains 22F, 39B, 39C, and 39F had a higher 7dCC, and 17A, 22H, 38A, and 38E obtained lower values (Figure 2C); however, only the 7dCC values of strains 22H and 38A differed significantly from that of 39F. It would seem that the maximum cell concentration of the yeasts with higher $\mu_{max}$ was reached on the first 3–4 days, before slightly lowering later. Thus, they showed a lower 7dCC than those with lower $\mu_{max}$.

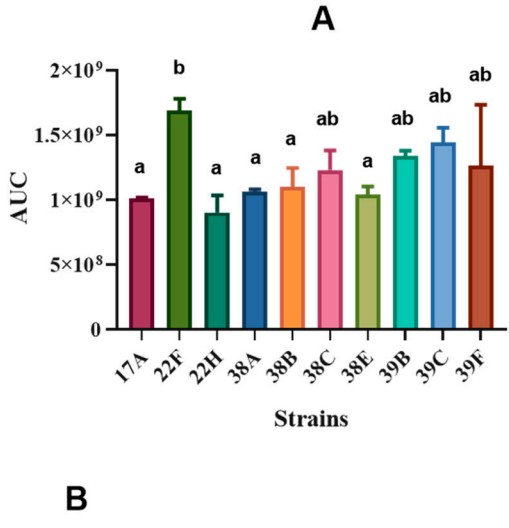

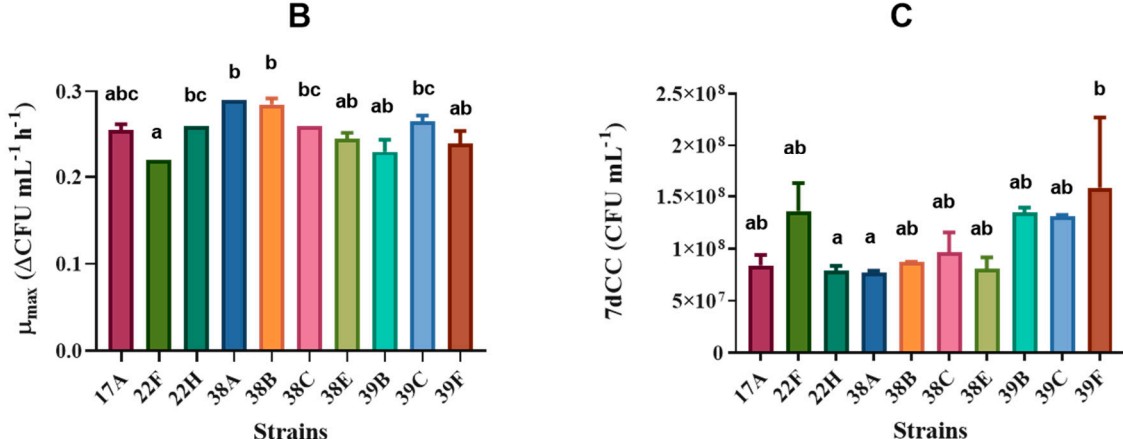

**Figure 2.** Growth parameters recorded for the different *S. cerevisiae* strains grown in sterile grape Garnacha must. (**A**): Area Under the Curve (AUC) calculated from the growth kinetics data; (**B**): the maximum growth rate ($\mu_{max}$) expressed as $\Delta$ CFU mL$^{-1}$ h$^{-1}$; (**C**): 7-day cell concentration (7dCC) expressed as CFU mL$^{-1}$ achieved at 7 days of fermentation. Different letters in the columns mean a significant difference ($p < 0.01$) between the values.

Between days 7 and 14 of AF, the viable cell concentration of strain 39F dropped by almost 70% (Figure S1). Less marked decreases were observed for strains 22H, 38B, and 39B (40–45%), whereas the population of the others diminished by less than 20%, or even remained unchanged (Figure S1). Some facts can partially explain the dominance relations between yeast strains in industrial winemaking, such as the different growth rates and

overall growth dynamics observed when strains grew individually in Garnacha grape musts. Two of the four strains that grew faster were dominant at HAF and FAF: 22H and 38A. The concentration of strain 22H lowered from 75% to 33% between HAF and FAF, which would explain the degree of dominance. Strain 38A only appeared at FAF and at a similar frequency to 22H (27% and 33%, respectively). Thus, besides 22H, this was why 38A co-led AF during the final moments of the fermentation.

The amount and kinetics of the consumed sugars, the produced ethanol, glycerol, and acetic acid, and ethanol yield were determined for each yeast strain. The glucose and fructose concentrations in must before inoculation were 120 and 122 g/L, respectively. The amount of glucose consumed on day 14 was similar for all the strains; however, significant differences were found between the group consisting of 17A, 22H, and 38B, and the rest. Glucose depletion was almost complete (Figure 3A), with residual concentrations remaining, which ranged from 0.15 to 2.60 g/L. Although all the strains efficiently degraded glucose, their consumption kinetics differed, as observed in Supplementary Figure S2A. Strains 38E and 39F showed quicker degradation kinetics and almost complete glucose uptake at 7 days, whereas strain 17A degraded this sugar more slowly and incompletely during the same time period. Strains 22H and 39C displayed intermediate growth rates and glucose consumption kinetics (Figure 2A and Supplementary Figure S2A). Differences in glucose uptake by several *S. cerevisiae* have also been observed by other authors, which could be the consequence of distinct expression levels, and of the activities of hexose transport and phosphorylation enzymes, and those of glycolytic and ethanol pathway enzymes [55–57].

The residual fructose concentrations on day 14 were higher than those of glucose (from 5 to 24 g/L), and yeasts showed different abilities to consume this sugar (Figure 3B); the strains that degraded fructose more rapidly and efficiently were 38E, 39B, 39C, and 39F, whereas 17A and 22H were slower and less efficient, and showed significant differences to the rest of the strains (Figure 3B and Supplementary Figure S2B). The strains showing the highest fructose consumption rates were 38E and 39F, whereas the lowest rates were for strains 17A and 22H (Supplementary Figure S1B). *S. cerevisiae's* preference for glucose instead of fructose is a fact that has been reported by many authors [58,59]. This fact is related to the kinetic constants (Km and Vmax) of "low affinity" and "high affinity", and to hexose transporters and hexose phosphorylating enzymes [59]. These activities are strain-dependent [55]. The discrepancy between glucose and fructose utilisation increases during fermentation, and depends on the chemical composition of the fermenting must. Thus, the increase in ethanol noted during AF had a stronger inhibitory effect on fructose than on glucose uptake, whereas the nitrogen supplementation of must improves fructose consumption more than glucose consumption [55].

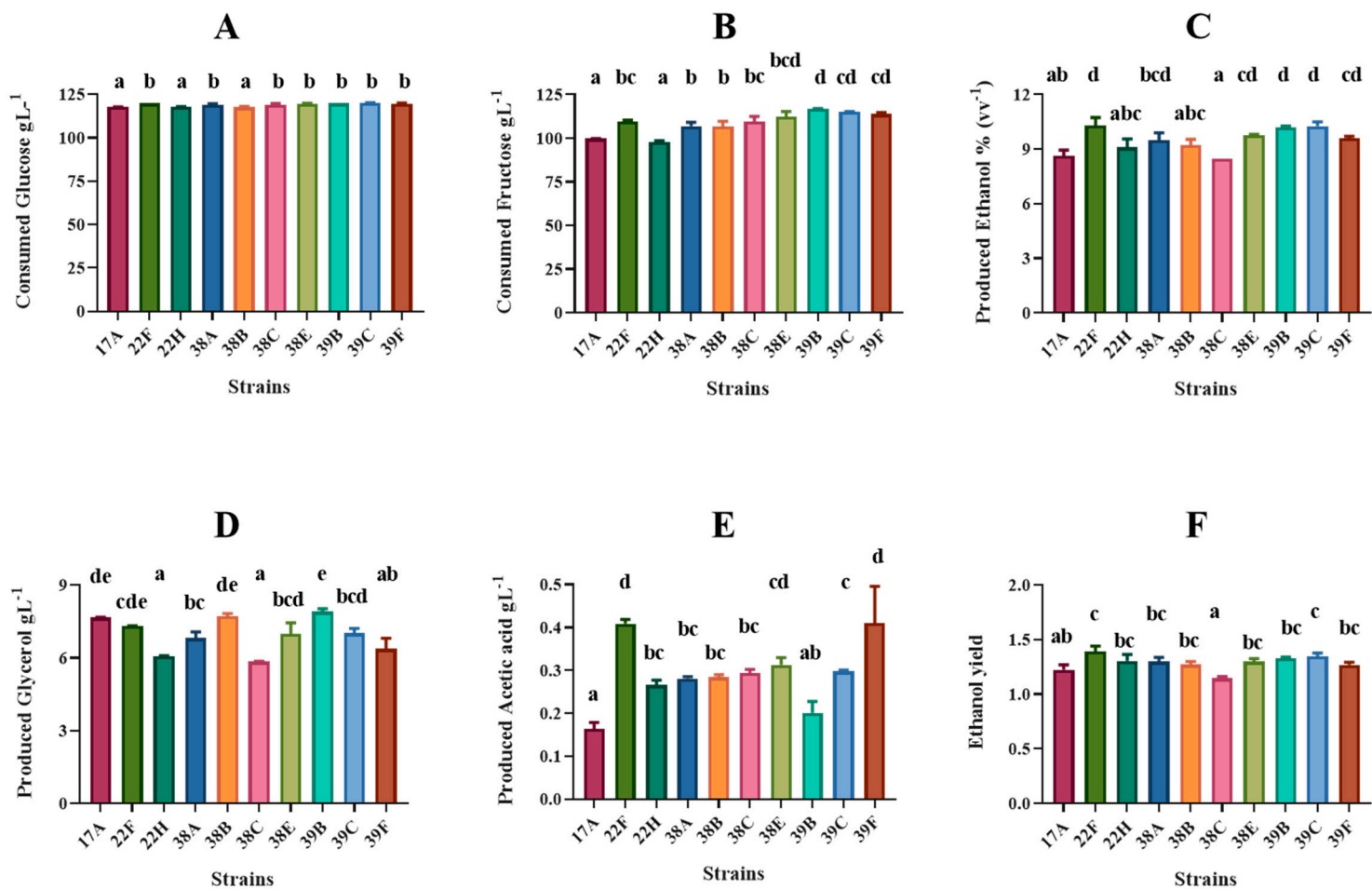

**Figure 3.** Consumed sugars, products generated, and ethanol yield from sugars on day 14 from the start of yeast inoculation. Glucose consumption expressed as g/L (**A**), fructose consumption expressed as g/L (**B**), ethanol production expressed as % (*v/v*) (**C**), glycerol production expressed as g/L (**D**), acetic acid production expressed as g/L (**E**), and ethanol yield (**F**). Different letters in the columns mean a significant difference (*p* < 0.01) between the values.

The ethanol produced on day 14 ranged between 8.5 and 10.5% (*v/v*). The strains that produced more ethanol were 22F, 39B, and 39C. Significant differences were found between the group consisting of 17A, 22H, 38B, 38C, and the group containing the strains 22F, 39E, 39C, and 39F. The strains that displayed the highest initial rates of ethanol production were 38A, 38E, and 39C. However, the ethanol amounts produced by 38A and 38E at the end of the experiment were not the highest because their ethanol production rates between days 7 and 14 were lower than those of other initially slower ethanol producers, such as 39B and 22F (Figure 3C and Supplementary Figure S2C). The strains producing higher glycerol contents were 17A, 38B, and 39B, with strains 22H and 38C yielding significantly less (Figure 3D). Glycerol production kinetics show that this compound was produced mainly on the first three days of fermentation (Supplementary Figure S2D). This is because most of the pyruvate from glycolysis is used for the biosynthesis of the molecules needed for active growth and an NAD+ deficit is generated, which must be recovered through the glyceropyruvic pathway [60]. The glycerol production kinetics ranked strains similarly to the ethanol production kinetics (Supplementary Figure S2C,D), which suggests that both productions are correlated. This fact does not support the results found by Quirós et al. [61], who reported that glycerol and ethanol were negatively correlated. Most strains generated acetic acid levels below 0.32 g/L, and only strains 22F and 39F produced more than 0.4 g/L of this acid (Figure 3E), showing significant differences to strain 17A. Five behaviours were displayed in relation to acetic acid production kinetics: (a) some strains began production after inoculation, but produced 0.2 g/L or less (strains 17A and 39B); (b) strains began production after three days and produced acetic acid concentrations ranging between 0.2 and 0.3 g/L (22H and 38B; (c) strains began synthesis between one and three days and produced less than 0.3 g/L (38A); (d) strains started synthesis after inoculation and produced 0.29–0.31g/L (38C, 38E and 39C); (e) strains began production after inoculation and yielded 0.4 g/L acetic acid or more (22F and 39F). These different behaviours were deduced from their distinct production kinetics (Supplementary Figure S2E). Differences in acetic acid production were possibly related to strains' distinct acetyl-CoA synthetase capacities. Indeed, the poor activity of this enzyme causes acetate overflow [59]. The molar ethanol yields are shown in Figure 3F. As we can see, the strains that less efficiently transformed sugars into ethanol were 17A and 38C (yield values of 1.26 and 1.15, respectively), whereas the most efficient ones were 22F and 39C (yield values of 1.39 and 1.36, respectively), which significantly differed from 17A and 38C. The other strains gave yields between 1.27 and 1.33.

From a global overview of strains' growth and metabolic characteristics, and considering those more important from an oenological point of view (high growth rates, good but not excessive ethanol yields, and low acetic acid production), strains 22 H, 38A, 38B, and 39C are good candidates to be used for "Pago" industrial Garnacha fermentations in the studied winery.

*3.3. Chemical Characteristics of Small-Scale Produced Wines*

Table 2 shows the mean values, standard deviations, and the ANOVA of the physico-chemical parameters of the wines produced by the small-scale vinification with the tested yeasts. Significant differences appear in all the analysed parameters (*p* value < 0.05), except for density because all the wines perfectly completed AF. All the tested yeasts completely consumed sugars (between 1.7 and 2.4 g/L). Thus, from the fermentation kinetics point of view, they all behaved in the same way by fulfilling the first requirement to select a yeast [62]. As volatile acidity was very low (from 0.17 to 0.37) in all the wines, all the studied strains are of interest in relation to this parameter [63]. The wines with the lowest pH (3.57) and high titratable acidity (6.15) were those fermented with strain 22H. Low pH values diminish wines' physico-chemical and microbial alterations [64]. The alcohol yield obtained for all the strains was similar, but the wines fermented with strain 22H had the highest alcohol content, followed by those fermented by strains 38A and 39F.

**Table 2.** Effect of fermentation with selected yeast on the sugar consumed during fermentation, the consumed sugar/ethanol production ratio and the Garnacha wine composition.

| Strain | Sugar Consumed | Consumed Sugar/Ethanol Production Ratio | Residual Sugar | Alcohol Concentration (%*v/v*) | Density (g/mL) | Volatile Acidity (g acetic acid/L) | pH | Titratable Acidity (g tartaric acid/L) |
|---|---|---|---|---|---|---|---|---|
| 17A | 193.00 ± 0.13 a | 17.71 ± 0.33 b | 2.1 ± 0.12 a | 10.91 ± 0.60 a | 993 ± 2.00 | 0.17 ± 0.13 a | 3.61 ± 0.18 ab | 6.1 ± 0.31 fg |
| 22F | 204.33 ± 1.53 b | 17.42 ± 0.24 a | 2.4 ± 0.23 a | 11.73 ± 0.15 bcd | 991 ± 0.00 | 0.37 ± 0.04 gh | 3.63 ± 0.04 ab | 5.73 ± 0.09 cd |
| 22H | 210.67 ± 2.52 b | 17.61 ± 0.26 b | 1.9 ± 0.18 a | 11.97 ± 0.12 d | 991 ± 0.00 | 0.24 ± 0.08 bc | 3.57 ± 0.07 a | 6.15 ± 0.15 ef |
| 38A | 211.67 ± 1.53 b | 17.69 ± 0.23 b | 2.2 ± 0.22 a | 11.93 ± 0.18 d | 991 ± 1.00 | 0.36 ± 0.04 fg | 3.76 ± 0.14 ab | 5.69 ± 0.29 def |
| 38B | 207.00 ± 2.00 b | 17.86 ± 0.73 c | 1.9 ± 0.25 a | 11.61 ± 0.36 bcd | 992 ± 2.00 | 0.24 ± 0.02 cd | 3.78 ± 0.02 c | 5.28 ± 0.04 ab |
| 38C | 204.00 ± 1.00 b | 17.44 ± 0.09 a | 2.3 ± 0.15 a | 11.71 ± 0.0 bcd | 991 ± 0.00 | 0.37 ± 0.0 gh | 3.8 ± 0.02 c | 5.2 ± 0.16 a |
| 38E | 204.67 ± 2.08 b | 17.80 ± 0.37 c | 1.7 ± 0.33 a | 11.51 ± 0.26 bc | 991 ± 0.00 | 0.29 ± 0.01 ef | 3.59 ± 0.07 a | 5.75 ± 0.16 d |
| 39B | 207.00 ± 2.00 b | 17.80 ± 0.55 c | 1.9 ± 0.09 a | 11.63 ± 0.25 bcd | 991 ± 0.00 | 0.18 ± 0.02 ab | 3.67 ± 0.02 b | 5.8 ± 0.16 de |
| 39C | 202.00 ± 2.65 b | 17.89 ± 0.48 c | 2.2 ± 0.16 a | 11.31 ± 0.36 ab | 991 ± 0.00 | 0.27 ± 0.02 de | 3.79 ± 0.03 c | 6.25 ± 0.11 g |
| 39F | 206.67 ± 3.52 b | 17.37 ± 0.50 a | 2.2 ± 0.31 a | 11.88 ± 0.15 cd | 991 ± 0.00 | 0.37 ± 0.05 h | 3.78 ± 0.1 d | 5.71 ± 0.22 bc |
| F-Ratio | 0.69 | 15.6 | 0.66 | 54.97 | 1.95 | 34.92 | 18.24 | 18.2 |
| *p*-Value | 0.4831 | 0.021 | 0.07312 | 0.0014 | 0.055 | 0 | 0 | 0 |

Different letters in the same column mean a significant difference ($p < 0.05$) in the fermented wines.

Different studies reported on the selection of *S. cerevisiae* yeasts from different varieties of grape must, and aimed to choose the most appropriate strains [65–67]. However, the present study allows more consistent results to be obtained because the same grape must was used to inoculate the different strains.

### 3.4. Colour and Tannins-Related Parameters

Table 3 shows the results obtained for the colour-related parameters of wines 1 month after AF was completed with the different yeasts, and that strains 38C and 22H were those that best maintained wine colour intensity. These strains also had the highest concentration of non-discolourable anthocyanins, and the wines made with strain 22H had the highest total anthocyanins concentration. This higher anthocyanin concentration could also be related to the lower polyphenol adsorption capacity of yeast cell walls [24]. The adsorption capacity of anthocyanins and tannins possesses a yeast strain-dependent character [19,68,69] and is related to the biomass, membrane composition, and cell wall/membrane integrity of each strain [16,70,71]. The ability to stabilise colouring matter after fermentation may be influenced by yeast strains because they influence the synthesis of carbonyl compounds (mainly acetaldehyde and pyruvic acid) that act as precursors for more stable molecules to form against $SO_2$ discolouration [20,69]. The anthocyanins that cannot be discoloured by $SO_2$ are also stable, as a result of their colour variation according to pH. This is because they are combined with other polyphenols which make them less sensitive to oxidation and discoloration [72]. By fermenting with a strain that provides high colour intensity, and one that is accompanied by a high total anthocyanin concentration, of which the maximum possible is in a form that cannot be discoloured by sulphur, wine with a stabler colour is obtained over time. In Garnacha wines, strain 22H was the most favourable one for maintaining colour, and not only after fermentation, but also perhaps throughout wine conservation.

Table 3 shows the results obtained for the tannin composition of the wines made with the different yeast strains selected in Garnacha grapes. Both the polyphenols and tannins concentrations obtained low values in all the wines made. This was partly due to the variety's polyphenolic characteristics, and also because the temperatures reached during fermentation (22–23 °C) do not allow a similar tannin concentration to be obtained to that achieved with temperatures close to 28 °C, which was used during winemaking in the studied winery. The polyphenol and tannin concentrations were higher in the wines fermented with strains 22H and 39C, while the catechin concentration was lower because tannins are formed by the polymerisation of single catechin molecules. The drop in catechins results in a reduced wine bitterness [73,74]. In addition, the wines with a higher condensed or polymerised tannin concentration, such as those made with strains 22H, 38E, and 39C, tend to be less bitter and astringent [75,76]. The DMACH (dimethyl amino cinnamaldehyde) Index is a measure of the average degree of tannin polymerisation with an inverse reading [37]. The tannins with the highest degree of polymerisation appeared in the wines from strains 22H, 38A, and 39C. The condensation of tannins and polysaccharides improves wine quality because it reduces astringency and improves unctuousness. This condensation is measured with the Ethanol Index, and the higher the Ethanol Index, the higher the degree of polymerisation with polysaccharides. The wines with the most polysaccharide-bound tannins were those made with strains 22H and 39B, and they had the highest Ethanol Index value. This was due to the property of the yeasts that released polysaccharides from their cell wall to the medium by β-glucanase activity, and the structural constituents of yeast walls containing glucans, mannans, and mannoproteins [77], which can bind to tannins by means of their free radicals to reduce their astringency and increase their unctuousness and mouthfeel [73,78].

**Table 3.** Effect of fermentation with the selected yeasts on the phenolic compounds of Garnacha wines.

| Strain | Colour Intensity (CI) | Hue | Coloured Anthocyanins (mg/L) | Total Anthocyanins (mg/L) | Catechins (g/L) | Condensed Tannins (g/L) | Total Phenolic Compounds (g/L) | Total Polyphenol Index | DMACH Index (%) | Ethanol Index (%) |
|---|---|---|---|---|---|---|---|---|---|---|
| 17A | 3.33 ± 0.23 a | 61.09 ± 0.59 c | 334.55 ± 27.28 ab | 457.5 ± 35.88 a | 0.11 ± 0.01 d | 0.762 ± 0.049 bc | 2.62 ± 0.11 ab | 29.75 ± 1.89 ab | 171.9 ± 20.24 cd | 52.21 ± 3.7 a |
| 22F | 4.88 ± 0.40 bcd | 55.25 ± 1.99 b | 342.41 ± 9.22 ab | 447 ± 7 a | 0.07 ± 0.01 a | 0.718 ± 0.099 a | 2.561 ± 0.07 ab | 29.42 ± 0.82 a | 191.05 ± 36.36 d | 55.4 ± 1.24 abcd |
| 22H | 5.75 ± 0.62 cd | 62.54 ± 4.45 c | 398.99 ± 18.07 d | 522.5 ± 50.61 c | 0.07 ± 0.01 abc | 0.869 ± 0.083 d | 2.77 ± 0.18 cd | 33.49 ± 2.19 ef | 136.16 ± 24.75 d | 59.78 ± 4.86 de |
| 38A | 5.23 ± 0.87 d | 52.7 ± 3.22 a | 364.4 ± 45.2 ab | 472 ± 57.34 a | 0.08 ± 0.0081 | 0.861 ± 0.159 abc | 2.74 ± 0.18 ab | 33.19 ± 3.08 abc | 146.66 ± 29.72 bcd | 58.04 ± 3.03 de |
| 38B | 4.07 ± 0.62 b | 55.43 ± 4.43 b | 361.54 ± 35.4 a | 416.5 ± 20.67 abc | 0.08 ± 0.00 bc | 0.71 ± 0.085 ab | 2.45 ± 0.08 a | 29.17 ± 1.3 a | 180.82 ± 24.32 cd | 53.09 ± 4.04 ab |
| 38C | 6.22 ± 0.7 d | 54.78 ± 0.84 b | 398.83 ± 58.03 ab | 449.8 ± 39.18 c | 0.09 ± 0.01 bc | 0.779 ± 0.034 bc | 2.67 ± 0.53 d | 31.32 ± 1.6 bcd | 188.43 ± 31.63 d | 53.92 ± 2.25 abc |
| 38E | 5.18 ± 0.45 cd | 53.69 ± 2.31 ab | 353.13 ± 29.14 ab | 441.9 ± 47.73 ab | 0.08 ± 0.01 bc | 0.886 ± 0.058 d | 2.76 ± 0.12 bcd | 31.9 ± 0.54 cde | 134.47 ± 25.66 a | 56.26 ± 3.43 bcde |
| 39B | 4.33 ± 0.54 bc | 54.54 ± 0.88 b | 327.19 ± 26.41 ab | 428.9 ± 39.81 a | 0.08 ± 0.01 bc | 0.793 ± 0.07 s | 2.47 ± 0.18 a | 30.77 ± 1.65 abc | 142.62 ± 10.69 ab | 59.53 ± 1.4 d |
| 39C | 4.69 ± 0.36 cd | 55.24 ± 1.62 b | 346.01 ± 38.19 bc | 473.5 ± 41.93 a | 0.08 ± 0.01 c | 0.878 ± 0.103 d | 2.71 ± 0.15 bc | 33.84 ± 1.67 c | 155.06 ± 13.37 abc | 57.2 ± 2.39 cde |
| 39F | 5.34 ± 0.55 bc | 53.12 ± 2.44 ab | 368.47 ± 25.27 cd | 477.8 ± 45.85 bc | 0.08 ± 0.01 bc | 0.855 ± 0.132 d | 2.71 ± 0.12 cd | 32.98 ± 2.87 def | 155.56 ± 21.76 a | 58.74 ± 4.31 de |
| F-Ratio | 15.09 | 9.75 | 4.35 | 3.88 | 10.7 | 9.65 | 3.52 | 5.01 | 4.58 | 4.36 |
| *p*-Value | 0 | 0 | 0.0003 | 0.0009 | 0 | 0.0002 | 0.0026 | 0.0001 | 0.0003 | 0.0005 |

Different letters in the same column mean a significant difference (*p* < 0.05) in fermented wines. DMACH, dimethylaminocinnamaldehyde.

The obtained results showed that the different yeast strains employed to carry out AF influenced the flavanols concentration because both monomers and condensed tannins modify not only polyphenolic content, but also the state and stability of polyphenolic compounds [20]. Variation in the composition of phenolic compounds is due to the different activity of yeast strains, the varying ability to extract phenolic compounds from grape skins, the distinct capacity to adsorb tannins or coloured compounds in their cell walls, and diverse metabolic or enzymatic activities [79–82]. Of all the tested strains, 22H maintained wine colour, and the concentrations of not only total and non-decolourisable anthocyanins, but also of polyphenols and tannins, by keeping them in a more polymerised state among them all and with polysaccharides.

### 3.5. Volatile Aroma Analysis

The contribution of the fermentation process to aroma has been studied from different perspectives: identifying the most influential variables by comparing aroma yields of different strains in a specific must [13,83]; and by studying yeast interactions during AF [3]. Among fermentative aromas, a distinction is made between the aroma compounds entirely synthesised by yeast as a result of the yeast metabolism, and the aromas generated by aromatic compounds being released from the non-aromatic precursors present in must by the action of yeast enzymes. The type and concentration of these precursors depend mainly on the grape variety used to make wine, which is why they also form part of the varietal aroma. The main types of fermentative aromas synthesised by yeasts are organic acids, higher alcohols, and esters, and to a lesser extent, aldehydes. Fermentative aromas can provide both positive sensations, such as fruity or floral aromas (esters and higher alcohols), and negative ones.

Twenty-five volatile compounds derived from yeast metabolism and belonging to five chemical families were determined in the wines fermented with the different yeasts: acetaldehyde, methyl acetate, diacetal, ethyl acetate, isobutyl acetate, ethyl isobutyrate, ethyl isovalerate, 1 butanol, isoamyl alcohol, ethyl octanoate, 1 heptanol, isobutyric, 5 methylfurfural, 2,3 butanediol, butyric acid, gamma butyrolactone, ethyl decanoate, isopentanoic acid, diethyl succinate, 2-phenylethyl acetate, hexanoic acid, benzyl alcohol, 2-phenyl ethanol, 2-ethyl hexanoic acid, octanoic acid, and decanoic acid. The remaining analysed compounds were 1 propanol, ethyl hexanoate, hexyl acetate, ethyl lactate, cis 3 hexenol, benzaldehyde, and diethyl glutarate. As they were not representative, they were not included in the statistical analyses. As shown in Table 4, yeast strain had a significant effect on the concentration of most of the volatile compounds in the Garnacha red wines.

Different studies have revealed that the contribution to wine aroma is made by odorant families rather than by individual compounds. The effect of each component of an aroma family is additive or synergistic so insofar as their aroma value is less than 1, even on an individual basis, and the total sometimes clearly exceeds it. As the aroma of the compounds of the same family is normally equal or similar, and differs from the base aroma, this means that the family's characteristic aromatic note can be perceived in wine. Therefore, to better analyse the effect of isolated indigenous yeasts, a study has been carried out on the different families of aromatic compounds [84].

**Table 4.** Effect of fermentation with selected yeast strains on the aromatic compounds of Garnacha wines.

| Volatile Compounds | Concentration (mg/L) | | | | | | | | | F-Ratio | *p*-Value |
|---|---|---|---|---|---|---|---|---|---|---|---|
| | 17A | 22F | 22H | 38A | 38B | 38C | 38E | 39B | 39C | | |
| Isoamyl alcohol | 57.66 ± 7.60 e | 50.12 ± 18.59 de | 36.13 ± 18.17 bcd | 25.36 ± 15.66 ab | 30.98 ± 2.59 abc | 36.70 ± 19.99 bcd | 37.18 ± 12.78 bcd | 19.41 ± 3.73 a | 29.51 ± 6.91 abc | 13.87 | 0.0000 |
| 2.3 Butanediol | 0.02 ± 0.30 abc | 0.024 ± 0.01 abc | 0.01 ± 0.00 a | 0.03 ± 0.00 cd | 0.07 ± 0.02 e | 0.03 ± 0.00 bc | 0.02 ± 0.01 abc | 0.02 ± 0.00 ab | 0.03 ± 0.01 cd | 78.73 | 0.0000 |
| 1-Heptanol | 0.041 ± 0.01 a | 0.09 ± 0.02 c | 0.15 ± 0.01 d | 0.10 ± 0.03 c | 0.05 ± 0.05 ab | 0.04 ± 0.04 a | 0.08 ± 0.00 bc | 0.08 ± 0.00 bc | 0.08 ± 0.00 bc | 10.50 | 0.0000 |
| Benzyl alcohol | 0.03 ± 0.01 ab | 0.03 ± 0.02 ab | 0.04 ± 0.00 bc | 0.04 ± 0.01 ab | 0.07 ± 0.02 d | 0.035 ± 0.015 ab | 0.03 ± 0.01 a | 0.06 ± 0.01 cd | 20.55 | 0.0000 |
| 2-Phenylethanol | 16.99 ± 2.58 c | 14.10 ± 5.94 bc | 14.41 ± 5.25 bc | 7.39 ± 2.89 a | 12.24 ± 1.72 ab | 12.71 ± 8.30 b | 10.80 ± 2.95 ab | 7.43 ± 0.78 a | 12.08 ± 4.48 ab | 10.54 | 0.0000 |
| Total alcohols | 74.75 | 64.38 | 50.75 | 32.93 | 43.39 | 49.87 | 48.12 | 26.96 | 41.76 | | |
| Methyl acetate | 0.06 ± 0.02 ab | 0.08 ± 0.05 ab | 0.16 ± 0.13 cd | 0.08 ± 0.05 ab | 0.17 ± 0.03 d | 0.03 ± 0.01 a | 0.08 ± 0.03 ab | 0.13 ± 0.02 bcd | 0.09 ± 0.07 abc | 40.56 | 0.0000 |
| Ethyl acetate | 0.07 ± 0.09 a | 0.04 ± 0.01 a | 0.02 ± 0.01 a | 0.23 ± 0.08 b | 0.37 ± 0.08 c | 0.04 ± 0.01 a | 0.02 ± 0.01 a | 0.19 ± 0.13 b | 0.03 ± 0.01 a | 18.67 | 0.0000 |
| Isobutyl acetate | 0.03 ± 0.00 a | nd | 0.05 ± 0.01 a | 0.05 ± 0.01 a | 0.19 ± 0.28 b | 0.03 ± 0.00 a | 0.03 ± 0.00 a | 0.02 ± 0.03 a | 0.03 ± 0.00 a | 4.57 | 0.0164 |
| Ethyl isobutyrate | 0.03 ± 0.01 abc | 0.06 ± 0.01 bc | 0.26 ± 0.09 d | 0.03 ± 0.01 abc | 0.02 ± 0.00 a | 0.07 ± 0.02 c | 0.05 ± 0.03 abc | 0.05 ± 0.01 abc | 0.03 ± 0.00 ab | 17.87 | 0.0000 |
| Hexyl acetate | 0.01 ± 0.00 a | 0.03 ± 0.00 de | 0.02 ± 0.01 c | 0.03 ± 0.01 d | 0.03 ± 0.00 de | 0.01 ± 0.00 ab | 0.02 ± 0.00 bc | 0.04 ± 0.01 f | 0.03 ± 0.00 de | 26.87 | 0.0000 |
| Ethyl octanoate | 0.19 ± 0.06 a | 0.45 ± 0.12 cd | 0.33 ± 0.21 abc | 0.35 ± 0.15 abc | 0.36 ± 0.16 bcd | 0.40 ± 0.17 bcd | 0.26 ± 0.06 ab | 0.20 ± 0.05 a | 0.52 ± 0.13 d | 43.56 | 0.0000 |
| Ethyl decanoate | 0.24 ± 0.07 bcd | 0.20 ± 0.08 abc | 0.34 ± 0.09 e | 0.13 ± 0.08 a | 0.30 ± 0.05 de | 0.17 ± 0.04 ab | 0.26 ± 0.05 cd | 0.23 ± 0.02 bcd | 0.25 ± 0.02 cd | 78.98 | 0.0000 |
| Diethyl succynate | 0.15 ± 0.02 bc | 0.19 ± 0.02 c | 0.32 ± 0.04 d | 0.18 ± 0.05 c | 0.04 ± 0.02 a | 0.14 ± 0.02 bc | 0.11 ± 0.05 b | 0.02 ± 0.00 a | 0.18 ± 0.08 c | 20.34 | 0.0000 |
| 2-Phenylethylacetate | 0.59 ± 0.28 ab | 0.55 ± 0.30 a | 0.88 ± 0.20 c | 0.60 ± 0.21 ab | 0.80 ± 0.17 bc | 0.55 ± 0.18 a | 0.63 ± 0.13 ab | 0.48 ± 0.09 a | 0.59 ± 0.14 ab | 16.78 | 0.0000 |
| Total esters | 1.38 | 1.6 | 2.39 | 1.68 | 2.3 | 1.44 | 1.46 | 1.18 | 1.72 | | |
| Butyric acid | 0.06 ± 0.04 ab | 0.11 ± 0.04 bc | 0.22 ± 0.08 e | 0.02 ± 0.01 a | 0.18 ± 0.03 e | 0.18 ± 0.04 de | 0.14 ± 0.03 cd | 0.11 ± 0.01 bc | 0.12 ± 0.01 c | 34.65 | 0.0000 |
| Isopentanoic acid | 0.55 ± 0.06 cd | 0.76 ± 0.13 e | 0.38 ± 0.18 b | 0.41 ± 0.17 bc | 0.39 ± 0.09 b | 0.35 ± 0.15 ab | 0.39 ± 0.12 b | 0.23 ± 0.03 a | 0.57 ± 0.10 d | 16.82 | 0.0000 |
| Hexanoic acid | 0.80 ± 0.14 bc | 0.53 ± 0.33 a | 1.09 ± 0.37 d | 0.63 ± 0.25 ab | 0.84 ± 0.23 cd | 0.91 ± 0.29 cd | 0.66 ± 0.12 abc | 0.47 ± 0.05 a | 1.08 ± 0.16 d | 21.45 | 0.0000 |
| 2-Ethyl Hexanoic acid | 0.01 ± 0.01 ab | 0.01 ± 0.01 ab | 0.05 ± 0.01 d | 0.02 ± 0.01 bc | 0.01 ± 0.00 ab | 0.02 ± 0.00 c | 0.01 ± 0.00 a | 0.01 ± 0.00 a | 0.00 ± 0.00 a | 6.87 | 0.0000 |
| Octanoic acid | 0.97 ± 0.25 bcd | 0.64 ± 0.29 ab | 0.84 ± 0.12 abc | 0.77 ± 0.32 abc | 0.82 ± 0.35 abc | 0.84 ± 0.15 abc | 1.01 ± 0.34 cd | 0.55 ± 0.08 a | 1.30 ± 0.38 d | 32.56 | 0.0000 |
| Decanoic acid | 0.23 ± 0.09 abc | 0.44 ± 0.14 e | 0.44 ± 0.12 bcde | 0.20 ± 0.06 ab | 0.40 ± 0.19 de | 0.28 ± 0.08 abcd | 0.23 ± 0.06 abc | 0.15 ± 0.04 a | 0.37 ± 0.20 cde | 14.58 | 0.0000 |
| Isobutyric acid | 0.29 ± 0.03 cd | 0.38 ± 0.03 d | 0.30 ± 0.15 cd | 0.23 ± 0.02 c | 0.02 ± 0.00 a | 0.37 ± 0.19 d | 0.18 ± 0.09 bc | 0.06 ± 0.03 a | 0.08 ± 0.12 ab | 13.78 | 0.0000 |
| Total acids | 2.92 | 2.86 | 3.34 | 2.3 | 2.67 | 2.94 | 2.64 | 1.58 | 3.55 | | |
| γ-butirolactona | 0.17 ± 0.07 a | 0.15 ± 0.03 a | 0.35 ± 0.15 cd | 0.16 ± 0.01 a | 0.34 ± 0.03 cd | 0.32 ± 0.14 bcd | 0.37 ± 0.20 d | 0.23 ± 0.19 abc | 0.18 ± 0.01 ab | 24.67 | 0.0000 |
| Total lactons | 0.17 | 0.16 | 0.35 | 0.16 | 0.34 | 0.32 | 0.37 | 0.24 | 0.18 | | |
| Acetaldehyde | 0.03 ± 0.01 a | 0.15 ± 0.02 bcd | 0.13 ± 0.07 bc | 0.21 ± 0.12 de | 0.13 ± 0.02 bc | 0.17 ± 0.07 cd | 0.099 ± 0.06 abc | 0.17 ± 0.03 cd | 0.28 ± 0.06 e | 25.44 | 0.0000 |
| Diacetyl | 0.01 ± 0.00 a | 0.02 ± 0.01 a | 0.04 ± 0.01 b | 0.04 ± 0.02 b | 0.03 ± 0.00 ab | 0.02 ± 0.00 a | 0.01 ± 0.02 a | 0.02 ± 0.00 a | 0.04 ± 0.02 b | 45.78 | 0.0000 |
| 5-Methylfurfural | 0.51 ± 0.08 c | 0.52 ± 0.12 c | 0.32 ± 0.15 ab | 0.32 ± 0.01 ab | 0.39 ± 0.05 abc | 0.45 ± 0.16 bc | 0.31 ± 0.11 ab | 0.26 ± 0.03 a | 0.52 ± 0.25 c | 21.65 | 0.0000 |
| Total aldehydes | 0.55 | 0.69 | 0.49 | 0.58 | 0.55 | 0.63 | 0.42 | 0.45 | 0.85 | | |

Different letters in the same row mean a significant difference (*p* < 0.05) in fermented wines. nd., not detected.

Regarding alcohols, the wines fermented with strain 17A (74.8 mg/L) had the highest concentration; those fermented with strain 22F ranked second (64.4 mg/L); while those fermented with strain 39B (27 mg/L) had the lowest concentration. Higher alcohols provide wines with vegetable and herbaceous notes, and are considered unpleasant. They can lead to negative sensations when they exceed 350 mg/L, and can mask the aromas provided by esters, or when they exceed their perception threshold [85]. It should be noted that this was not the case with any of the studied strains. Moreover, when the total alcohol concentration is below 300 mg/L, they provide a pleasant profile and contribute positively to wine aroma. This is because they are precursors (besides organic acids) of esters [86]. In alcohols, the most important volatile compound is 2-phenylethanol because it considerably contributes positively to the aromatic profile by conferring sweet and floral (rose and lilac) notes. The wines fermented with strains 17A (16.99 mg/L), 22H (14.41 mg/L), and 22F (14.10 mg/L) had the highest 2-phenylethanol contents.

Esters are the most important group of volatile compounds because they contribute positively and significantly to fruity and floral wine aromas [86,87]. A high concentration of esters in wine is a putative indicator of a fruity aroma, and it is believed that there are synergistic effects between the compounds of this chemical family [88]. The wines fermented with strain 22H (2.4 mg/L), followed by those fermented with strain 38B (2.3 mg/L), had the highest ester concentrations, while those fermented with strain 39B (1.2 mg/L) had the lowest concentration. Although not all esters are beneficial for quality, ethyl acetate and methyl acetate confer an unpleasant solvent aroma at high concentrations, which are considered a defect in wine. However, they provide fruity aromas at a low concentration. All the yeast strains produced low ethyl and methyl acetate concentrations in wines (0.03–0.17 mg/L). A higher concentration was found for both ethyl isobutyrate (apple, strawberry) and diethyl succinate (caramel) in the wines fermented with strain 22H. A higher ethyl octanoate concentration (pineapple pear, floral) appeared in the wines fermented with strains 22F, 22H, 38A, 38B, and 38C. Strains 22H and 38B conferred wines higher ethyl decanoate concentrations (fruity, honey). A higher concentration of 2-phenylethyl acetate, which is an ester that confers fruity, honey, and rose aromas to wine [89], was noted for the wines fermented with strains 22H and 38B.

γ-butyrolactone provides characteristic sweet, coconut, plum, and caramel aromas. The strains with the greatest capacity to synthesise this compound in Garnacha wines were 38E, 22H, 38B, and 38C. This yeast characteristic is important because γ-lactones improve aromatic complexity for being associated with lactic notes of red wines [84,90].

Fatty acids are described as cheesy and buttery rancid aromas, and are therefore considered unpleasant when their total exceeds 20 mg/L. However, they are desirable when their concentration is below their perception threshold because they contribute to wine complexity by means of esterifying with alcohols, which gives rise to fruity esters [87]. The wines fermented with strains 39C and 22H had the highest acid concentrations (3.6 and 3.3 mg/L). The aromatic influence of these compounds has not been extensively studied compared to ethyl esters, although some (hexanoic acid, octanoic acid, decanoic acid, and isovaleric acid) have been recently reported as chemical compounds with a strong aromatic impact on wine [91–93].

Aldehyde content in wine is believed positive when they are below their perception threshold, which was the case of the wines in this study, but negative if their concentration exceeds the threshold [85]. The most important compound is acetaldehyde, which is a pleasant aroma at low concentrations and confers wine fruity aromas [94]. The wines fermented with strains 39C and 38A had the higher concentrations of this compound. Diacetyl contributes by conferring dairy and buttery notes [95]. The higher concentrations of this compound were found in the wines fermented with strains 22H, 38A, and 39C. 5-methyl furfural is related to wine barrel ageing, but can also be synthesised or degraded by yeasts during fermentation [96]. Strains 22F, 38C, and 39C produced higher concentrations of this compound in wines.

From the obtained results, it can be deduced that yeast strain significantly influenced the aromatic composition of Garnacha wines. Yeast metabolism brings about differences in the concentration of higher alcohols, esters, fatty acids, and aldehydes [9]. Studying the ability of yeast to produce volatile compounds is necessary to select the most suitable strain [97].

The wines fermented with strain 22H had the highest total concentration of esters (ethyl isobutyrate, ethyl octanoate, ethyl decanoate, diethyl succinate, and phenylethylacetate), high concentration of acids (hexanoic, ethyl hexanoic, and decanoic) and of also γ-butyrolactone, and 2-phenylethanol. This strain did not produce detrimental concentrations for any compounds, and can be considered appropriate for obtaining wines with good aromatic quality.

### 3.6. Sensory Profile of Garnacha Wines

The wines' sensory profiles were determined by a comparative sensorial analysis of the wines fermented with the different yeasts to select the yeast/s that could improve wines' organoleptic characteristics.

Table 5 shows that some descriptors were significantly influenced by the yeast strain. The highest scoring wines were those fermented by strain 22H in terms of colour parameters (intensity and quality, 8.9 and 7.2 points, respectively), aroma intensity and quality (7.9 and 7.9), red fruit (5.9), and black fruit aromas (4.9), taste intensity and quality (7.7 and 8.2), and overall quality (8.7). The wines fermented by strains 38B and 39B scored high for aroma intensity and quality, red fruit aroma, and overall quality. No significant differences were observed in the intensity and quality of the colour of the wines fermented with the different yeasts, but significant differences in aroma intensity, aroma quality, and red fruit aroma were recorded in the wines fermented with yeasts 22H, 38B, and 39B, of which the first two had higher ester concentrations. The sensory analysis revealed that the best rated wines were those fermented with strain 22H based on good colour intensity and quality, higher intensity and quality of aroma, and a better overall quality score. The wines fermented with strains 38B and 39B were also well rated from an organoleptic point of view and could also be considered to improve the Garnacha wines of the Pago winery.

**Table 5.** Effect of fermentation with selected yeast strains on the sensory attributes of Garnacha wines.

| Sensory Attributes | Scale of 1–10 | | | | | | | | | F-Ratio | *p*-Value |
|---|---|---|---|---|---|---|---|---|---|---|---|
| | 17A | 22F | 22H | 38A | 38B | 38C | 38E | 39B | 39C | | |
| **Colour** | | | | | | | | | | | |
| Colour quality | 6.89 ± 1.2 a | 7.11 ± 1.5 a | 8.89 ± 1.6 a | 6.24 ± 1.9 a | 8.6 ± 1.9 a | 7.63 ± 1.1 a | 6.46 ± 1.58 a | 8.33 ± 1.5 a | 7.11 ± 1.8 a | 0.59 | 0.740 |
| Colour intensity | 6.22 ± 1.04 a | 6.78 ± 1.48 a | 7.17 ± 1.29 a | 6.34 ± 1.8 a | 7.00 ± 1.85 a | 7.00 ± 1.1 a | 6.59 ± 1.59 a | 7 ± 1.78 a | 6.44 ± 1.02 a | 0.30 | 0.940 |
| **Aroma** | | | | | | | | | | | |
| Aroma intensity | 6.78 ± 0.4 a | 7.67 ± 1.1 a | 7.9 ± 1.4 b | 5.98 ± 0.97 a | 7.22 ± 0.97 b | 7.38 ± 0.83 b | 6.07 ± 1.10 a | 7.11 ± 1.2 b | 6.89 ± 1.9 a | 4.30 | 0.009 |
| Aroma quality | 5.89 ± 1.05 ab | 7.22 ± 1.09 bc | 7.89 ± 0.92 c | 5.29 ± 1.1 a | 7.33 ± 0.87 c | 4.75 ± 0.89 a | 5.22 ± 1.58 a | 7.72 ± 1.53 bc | 5.56 ± 2.55 a | 5.52 | 0.005 |
| Red fruits | 3.56 ± 1.1 a | 6.00 ± 1.1 b | 5.89 ± 1.2 b | 4.57 ± 0.88 a | 4.22 ± 0.5 a | 2.75 ± 0.90 a | 3.44 ± 0.6 a | 5.67 ± 1.01 b | 3.75 ± 0.55 a | 3.75 | 0.013 |
| Black fruits | 3.00 ± 0.5 a | 3.67 ± 1.1 a | 4.89 ± 1.3 a | 3.12 ± 0.76 a | 4.22 ± 1.2 a | 2.38 ± 0.67 a | 3.08 ± 0.8 a | 4.56 ± 1.00 a | 3.13 ± 0.8 a | 0.50 | 0.810 |
| Floral | 1.44 ± 0.33 a | 1.56 ± 0.7 a | 2.44 ± 0.6 a | 2.12 ± 0.55 a | 1.67 ± 0.6 a | 1.38 ± 0.55 | 1.66 ± 0.50 a | 1.78 ± 0.9 a | 1.75 ± 0.7 a | 0.30 | 0.960 |
| Balsamic | 1.11 ± 0.3 a | 2.89 ± 0.6 a | 2.78 ± 0.7 a | 2.31 ± 0.61 a | 2.89 ± 0.7 a | 1.66 ± 0.43 | 1.32 ± 0.6 a | 2.56 ± 0.3 a | 1.5 ± 0.55 a | 1.02 | 0.420 |
| Spicy | 2.22 ± 0.1 a | 2.33 ± 0.3 a | 2.11 ± 0.33 | 2.44 ± 0.55 | 2.00 ± 1.9 a | 1.66 ± 0.76 a | 2.79 ± 0.2 a | 2.67 ± 0.43 a | 1.13 ± 0.3 a | 0.72 | 0.600 |
| Lactic | 1.00 ± 0.00 a | 1.00 ± 0.00 a | 1.00 ± 0.00 a | 1.06 ± 0.11 a | 1.00 ± 0.00 a | 0.88 ± 0.11 a | 1.74 ± 0.00 a | 1.00 ± 0.00 a | 1.13 ± 0.00 a | 1.10 | 0.370 |
| Vegetable | 2.44 ± 0.4 a | 1.56 ± 0.10 a | 1.56 ± 0.52 a | 2.57 ± 0.77 | 2.33 ± 3.11 a | 2.38 ± 0.97 | 2.62 ± 2.10 a | 2.22 ± 0.40 a | 2.50 ± 0.5 a | 0.44 | 0.840 |
| Aromatics herbs | 1.11 ± 0.33 a | 1.78 ± 1.33 a | 1.67 ± 1.33 a | 1.68 ± 0.43 a | 1.22 ± 1.3 a | 1.00 ± 0.33 | 1.3 ± 0.4 a | 1.67 ± 0.4 a | 1.38 ± 0.7 a | 0.52 | 0.590 |
| Chocolate | 1.11 ± 0.33 a | 1.33 ± 0.001 a | 1.89 ± 0.2 a | 1.42 ± 0.43 a | 1.00 ± 0.0 a | 1.38 ± 0.26 a | 1.21 ± 0.0 a | 1.33 ± 1.4 a | 1.00 ± 0.0 a | 0.97 | 0.450 |
| **Taste** | | | | | | | | | | | |
| Taste intensity | 6.11 ± 1.5 a | 7.33 ± 1.1 a | 7.67 ± 1.2 | 6.11 ± 1.1 a | 7.67 ± 1.5 a | 6.75 ± 1.1 a | 6.32 ± 1.2 a | 8.11 ± 0.78 a | 6.43 ± 1.5 a | 1.52 | 0.190 |
| Taste quality | 5.89 ± 1.2 a | 7.22 ± 1.4 a | 8.19 ± 2.3 a | 5.67 ± 1.4 a | 7.11 ± 1.9 a | 6.25 ± 1.5 a | 5.72 ± 1.3 a | 7.67 ± 1.2 a | 6.07 ± 1.9 a | 0.59 | 0.730 |
| Acidity | 5.00 ± 1.3 a | 7.67 ± 2.3 a | 7.56 ± 1.4 a | 5.36 ± 1.5 a | 7.78 ± 1.3 a | 5.88 ± 1.5 a | 5.16 ± 1.09 a | 6.89 ± 1.05 a | 6.29 ± 1.03 a | 2.74 | 0.061 |
| Sweetness | 2.33 ± 0.21 a | 2.33 ± 0.43 a | 2.44 ± 0.21 a | 2.78 ± 0.42 a | 2.22 ± 0.27 a | 2.5 ± 0.42 a | 2.88 ± 0.34 a | 2.22 ± 0.32 a | 2.86 ± 0.72 a | 0.22 | 0.970 |
| Unctuousness | 2.44 ± 0.55 a | 2.89 ± 0.43 a | 3.89 ± 0.53 a | 2.9 ± 1.03 a | 3.00 ± 0.76 a | 3.63 ± 0.53 a | 2.64 ± 0.43 a | 3.33 ± 0.43 a | 2.14 ± 0.32 a | 0.44 | 0.840 |
| Structure | 3.33 ± 0.33 a | 4.00 ± 0.55 a | 3.67 ± 0.44 a | 3.67 ± 0.44 a | 3.67 ± 0.44 a | 3.33 ± 0.35 a | 3.25 ± 0.37 a | 3.87 ± 0.67 a | 3.33 ± 0.43 a | 2.29 ± 0.55 a | 0.16 | 0.980 |
| Astringency | 2.56 ± 0.56 a | 3.67 ± 0.65 a | 3.67 ± 0.59 a | 2.9 ± 0.65 a | 3.44 ± 0.72 a | 0.76 ± 0.54 a | 2.98 ± 0.65 a | 3.56 ± 0.65 a | 2.43 ± 0.54 a | 0.36 | 0.900 |
| Bitterness | 2.89 ± 0.77 a | 4.00 ± 0.64 a | 4.11 ± 0.63 a | 2.68 ± 0.56 a | 3.44 ± 0.55 a | 3.63 ± 0.85 a | 2.77 ± 0.63 a | 4.00 ± 0.54 a | 3.00 ± 0.53 a | 0.17 | 0.980 |
| Taste persistence | 3.44 ± 0.43 a | 4.78 ± 0.53 a | 5.56 ± 0.44 a | 3.98 ± 0.43 a | 5.22 ± 1.02 a | 4.25 ± 0.98 a | 3.62 ± 0.65 a | 5.67 ± 0.78 a | 3.57 ± 0.54 a | 0.81 | 0.560 |
| Overall quality | 5.89 ± 0.56 a | 7.94 ± 0.76 ab | 8.7 ± 0.87 b | 6.12 ± 0.88 a | 8.39 ± 0.99 b | 6.44 ± 0.65 a | 6.26 ± 0.55 a | 8.67 ± 0.74 b | 6.71 ± 0.77 a | 10.55 | 0.001 |

Different letters in the same row mean a significant difference (*p* < 0.05) in fermented wines.

### 3.7. Multivariate Data Analysis of Garnacha Wines

A PCA of 27 wines and 62 variables was performed to correlate the physico-chemical parameters [6], the phenolic [10] and aromatic compounds' [23] concentrations, and the sensory parameters [23] of the wines fermented with the different yeasts. The biplot showed that the first two principal components (PCs) explained 99.7% of the variance (PC1 = 75.4% and PC2 = 24.3%) of the dataset (Figure 4). The scores plot shows the distribution of the yeast strains (Figure 4A), while the loading plot, which indicates the weight of the variables (Figure 4B), represents the arrangement of the different parameters on the plane formed by PC1 and PC2. In the scores graph (Figure 4A), PC1 allowed wines to be separated into three groups. On the left are strains 17A, 38A, and 38B, in the centre of the coordinates are 22F, 22H, 39B, and 39C, with yeast strains 38E and 38C on the right. The loading plot indicates that the wines fermented with strains 22F, 22H, 39B, and 39C are related to diethyl succinate, ethyl octanoate, 2-phenylethyl acetate, ethyl hexanoic acid, titratable acidity, alcohol degree, coloured anthocyanins, and total polyphenols.

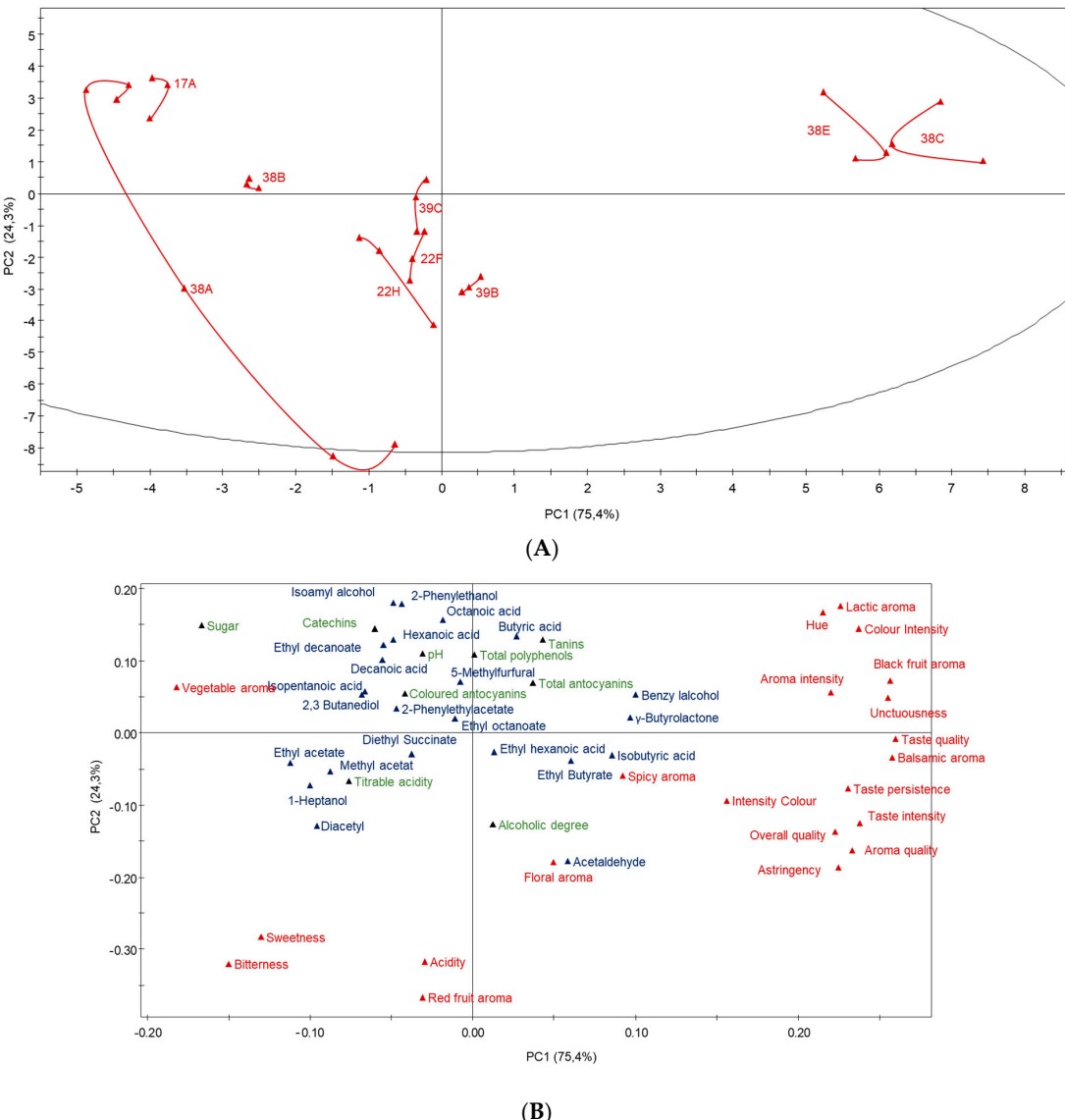

**Figure 4.** (**A**) Score plot and (**B**) loading plot on principal components PC1 and PC2 corresponding to the principal component analysis of the sensory characteristics (**A**) and chemical composition (**B**) of Garnacha wines fermented with different yeasts.

The wines fermented with strains 38C and 38E showed excellent sensory attributes: colour intensity, black fruit aroma, and unctuousness. The loading plot indicates that the wines fermented with strains 17A, 38A, and 38B were separate from the others based on their vegetable aroma, sweetness, bitterness, red fruit aroma, sugar, and their concentrations of diacetyl, ethyl acetate, 2–3 butanediol, and isopentanoic acid.

## 4. Conclusions

A high diversity of yeast species was found in the Pago Garnacha grape must; the more abundant species were *Hanseniaspora guillermondi*, *Metschnikowia pulcherrima*, and *Hanseniaspora opuntiae*. This diversity decreased as fermentation progressed, and only *S. cerevisiae* remained at FAF.

Ten different *S. cerevisiae* strains were identified and nine of them conducted AF.

*S. cerevisiae* strains showed differences in both growth parameters and fermentative behaviour.

The small-scale produced wines differed in terms of alcohol concentration, volatile acidity, pH and titratable acidity, colour-related phenolic compound, aroma-related compounds, and sensorial attributes. Differences were related to the employed *S. cerevisiae* strain to ferment the Pago Garnacha grape must.

The wines fermented with strain 22H obtained higher values for the colour-related parameters. The colour intensity of these wines was the highest and showed optimal total and non-discolourable anthocyanin concentrations. In addition, their total polyphenol and condensed tannin concentrations were high.

The wines fermented with strain 22H had high concentrations of esters, acids, γ-butirolactone, and 2-phenylethanol, which contribute to quality aroma.

Overall, strain 22H quickly grew, produced wines with moderate ethanol concentrations and low volatile acidity, and obtained the highest colour and aroma scores, plus a high score for sensory attributes. Although other strains stand out for some of these parameters, strain 22H obtained the best global characteristics.

**Supplementary Materials:** The following supporting information can be downloaded at https://www.mdpi.com/article/10.3390/beverages9010017/s1, Figure S1: Growth kinetics recorded for the different *S. cerevisiae* strains grown in sterile grape Garnacha for 14 days. Growth is recorded as Units forming Colonies per mL UFC/mL/; Figure S2: Glucose and fructose consumptions and ethanol, glycerol and acetic acid productions kinetics over 14 days from the beginning of yeast inoculation in sterile Garnacha grape must. Glucose consumption kinetics expressed as g/L (A), fructose consumption kinetics expressed as g/L (B), ethanol production kinetics expressed as % (*v*/*v*) (C), glycerol production kinetics expressed as g/L (D), acetic acid production kinetics expressed as g/L (E).

**Author Contributions:** Conceptualization, M.J.G.-E.; I.Á., I.P. and V.L.; methodology, I.Á., I.P., M.J.G.-E., V.L. and S.F.; validation, C.B., L.P., I.Á., I.P. and S.F.; formal analysis, C.B., I.P. and M.J.G.-E.; investigation, C.B., L.P., V.L. and M.J.G.-E.; writing—original draft preparation, C.B., I.P. and M.J.G.-E.; writing—review and editing, C.B., I.P. and M.J.G.-E.; visualization, I.Á., S.F. and M.J.G.-E.; supervision, I.Á., I.P., S.F. and M.J.G.-E.; project administration, I.Á. and I.P.; funding acquisition, I.Á. and I.P. All authors have read and agreed to the published version of the manuscript.

**Funding:** This work was supported by the Instituto Valenciano de Competitividad Empresarial (IVACE) (Ref: IFIDUA/2015/13).

**Institutional Review Board Statement:** Not applicable.

**Informed Consent Statement:** Not applicable.

**Data Availability Statement:** The data presented in this study are available upon request from the corresponding author.

**Acknowledgments:** The authors thank the winery Chozas-Carrascal (San Antonio de Requena, Spain) for providing samples, grape musts, and technical assistance.

**Conflicts of Interest:** The authors declare no conflict of interest. The funders have not intervened in the collection, analysis, and interpretation of the data, nor in the report writing, or the decision to publish the manuscript.

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
