# Peer review of "Influence of Native S. cerevisiae Strains on the Final Characteristics of “Pago” Garnacha Wines from East Spain"

_beverages, doi:10.3390/beverages9010017_

Round 1

Reviewer 1 Report

In their Manuscript «beverages-2143915» entitled "Influence of native S. cerevisiae strains on the final characteristics of “Pago” Garnacha wines from east Spain" for Beverages (journal), authors have reported somewhat original data and I think that the study is very interesting and it deserves to be published.

However, two major points and some minor points of criticism and questions have to be clarified prior publication:

MAJOR REVISION:

1 1) ABSTRACT: Why did you select yeast strain with high ethanol concentration? The present trend is trying to select low ethanol concentration strain, in order to fight against climate change (and global warming). This point must be absolutely clarified.

2 2) Mannoproteins are one of the major polysaccharide groups present in wine, having their origin in the yeast cell walls. Gonçalves, Heyraud, Pinho, and Rinaudo (2002) found that 32.2% of the total polysaccharides content of white wine corresponded to mannoproteins. The interaction between colour compounds and tannins with mannoproteins is extremely important as it can influence colour stability and improve sensory qualities. In my opinion, this point should be particularly necessary for a good study about the final characteristics of “Pago” Garnacha wines, in order to an adequate explanation about colour compounds content and behavior.  Therefore, I think some mannoproteins aspect should be included in the present study.

MINOR REVISION:

ABSTRACT: Saccharomyces cerevisiae must be always written in cursive.

LINE 52: Saccharomyces must be written in cursive. The same comment in the rest of the document.

LINE 62: I would suggest to write “…one of the factors which influences on the final wine quality…”. Otherwise, it seems that this is the only important factor in wine final quality.

LINE 74-81: This should be referenced in some way.

LINE 114: This information is not yet necessary, because it is previously given.

LINE 138: I don’t understand what it is really reference 25.

LINE 205: Some information about “Velcorin” should be given.

LINES 311-312: 1.8, 2.6 and 8.4, in order to an easier reading and to keep the same decimals number in relation to SD.

LINE 370: Concerning “…in white Gewurztraminer and red Bordeaux fermentations, respectively…”…Gewurtztraminer is a cultivar grape and Bordeaux, in my knowledge, is a region. This sentence seems strange to me and, in my opinion, should be modified.

Figure 2A it is not clear. In my opinion, it should be modified in any way (different form or different size).

LINE 423: 122 (in order to an easier reading).

LINE 432-435: Good hypothesis

Figure 3 (and Figure 2B, 2C and 2D): I would indicate the significance letters over the columns.

LINE 441 “…5 to 24…” (in order to an easier reading)

LINES 627-629: 74.8, 64.4 and 43.4 (in order to an easier reading)

LINES 646-648: 2.4, 2.3 and 1.2 (in order to an easier reading)

LINE 672: 3.6 and 3.3 (in order to an easier reading)

LINES 705-708: 8.9, 7.2, 7.9, 7.9, 5.9, 4.9, 7.7 and 8.2 (in order to an easier reading)

Reviewer 2 Report

The publication is very well structured and presents the entire stage of isolation and selection of yeasts for the production of high quality regional wines.
I congratulate the authors for their research.

I have some technical notes on the post:
1. All names of microorganisms must be Italics.
2. Some of the figures have a lower quality that needs to be improved.

Round 2

Reviewer 1 Report

In their Manuscript «beverages-2143915» entitled "Influence of native S. cerevisiae strains on the final characteristics of “Pago” Garnacha wines from east Spain" for Beverages (journal), authors have reported somewhat original data and I think that the study is very interesting and it deserves to be published.

Just in line 17, Saccharomyces cerevisiae should be written in cursive.